# OFFLINE_RL_OPE: A PYTHON PACKAGE FOR OFF-POLICY EVALUATION OF OFFLINE RL MODELS WITH REAL WORLD DATA

## ABSTRACT

offline_rl_ope is a fully unit tested and runtime type checked Python package for performing off-policy evaluation of offline RL models. offline_rl_ope has been designed for OPE workflows using real world data by: naturally handling uneven trajectory lengths; including novel convergence metrics which do not rely on OPE estimator ground truths; and providing a compute and data efficient API which can be integrated with many offline RL frameworks. This paper motivates and describes the core API design and functionality to enable ease of use and extension. The implementations of OPE methods have been benchmarked against existing implementations to ensure consistency and reproducibility. The offline_rl_ope source code can be found on GitHub at: REDACTED

## 1 INTRODUCTION

Offline RL enables MDPs to be solved without interaction with an environment (i.e., with only a logged (batch) dataset) and has grown in popularity recently due to the availability of such data and the challenges of performing environment interactions in high stakes settings (Levine et al., 2020). A core challenge however, when environment interaction is not possible (fully offline RL (ORL)) is off-policy evaluation (OPE). OPE refers to evaluating a hypothetical target policy, $\pi_e$ with access to trajectories generated according to an alternate policy, $\pi_\beta$. Since performing off-policy evaluations is inherently counterfactual, OPE must be performed carefully and is still an active research area.

There still does not exist a well established and tested code base for performing OPE. Such a codebase, which is agnostic to the implementation of the policy learning algorithm, would be beneficial to ensure reproducibility and transparency in the application of OPE. This ambiguity in the application of OPE estimators has been rooted in the non-uniqueness of estimators (e.g., weighted Per-Decision proposed by Precup et al. (2000) and Kallus & Uehara (2019)) and the use of custom implementations of estimators without declaration of changes (e.g., value clipping by Raghu et al. (2017)).

offline_rl_ope, is a unit tested and runtime type-checked Python library for performing off-policy evaluation on real world data, which is agnostic to the framework used for training ORL models. Specifically, the contributions are as follows:

- Developed for use with real world data: handles uneven trajectory lengths (section 3.1); and includes offline OPE evaluation metrics (section 3.3.3) and common techniques (e.g., clipping) for importance sampling (IS) and doubly robust (DR) estimators (section 3.1);
- An API that can be easily extended for research purposes or used as plug-and-play;
- Optionally integrates with d3rlpy (Seno & Imai, 2021) for train-time evaluations (as a posed to post-hoc).

The focus for the first release of offline_rl_ope has been to implement standard techniques for IS and DR in a flexible and efficient API. The first release has focused primarily on IS and DR estimates due to the existence of a strong implementations of FQE by Seno & Imai (2021). More advanced

IS methods (i.e., marginal state IS), model-based methods, composite methods, direct methods and efficient influence function methods will be included in future releases. The package code base is available at: REDACTED

The aim of this paper was to: introduce the API, enabling future work and use of offline_rl_ope; and to motivate many of the API structure decisions and offline metrics. This document was written with respect to version 7.0.1 of offline_rl_ope and accompanying code can be found at: REDACTED. API usage examples are provided in appendix C and at REDACTED.

## 2 RELATED WORK

Existing OPE codebases have generally been included within a larger RL framework (Kiyohara et al. (2023), Liang et al. (2018),Kiyohara et al. (2023)) and as a result, the OPE API is tightly coupled with a specific model training framework.

### 2.1 SCOPE-RL

Scope-RL (Kiyohara et al. (2023)) was the first library to focus predominantly on OPE and is the existing work that is most similar to offline_rl_ope. However, there exist a number of areas of divergence between Scope-RL and offline_rl_ope, the most critical of these being audience as offline_rl_ope is more appropriate for real world workflows whilst Scope-RL is tailored more towards research. This along with other differentiating factors have been described in table 1 and are described in greater detail in appendix A.

Table 1: Comparison of offline_rl_ope against Scope-RL

|  | **offline_rl_ope** | **Scope RL** |
|---|---|---|
| **Audience** | Real world analysis | OPE research |
|  | ORL framework agnostic | Deep integration with d3rlpy |
|  | Train time evaluations (w. d3rlpy) | ✗ |
|  | Uneven trajectory lengths | ✗ |
|  | Non-oracle metrics | Oracle metrics |
|  | Propensity modelling | Oracle behaviour policy |
|  | Framework agnostic OPE pipeline | End-to-end ORL (w. d3rlpy) & OPE pipeline |
| **Estimators** | Basic estimators | Basic and advanced estimators |
| **API design** | Extendable through equ. 1 | Limited extendability |
| **Continuous action spaces** | Stochastic policies only | Kernel smoothing of actions (Kallus & Zhou (2018)) |

## 3 HIERARCHY OF IS METHODS

Uehara et al. (2022) provides an overview of OPE methods for ORL, however, introduced below, is a 'hierachy of IS methods' which was critical in the design of the offline_rl_ope IS API (including importance sampling for DR). IS estimators have predominantly suffered from high variance, and as such a large amount of research has been dedicated to reducing it (Kallus & Uehara (2019), Thomas & Brunskill (2016), Precup et al. (2000)). The line of research broadly aligns to utilising control estimators from Monte Carlo statistics (Robert & Casella (2004), Thomas & Brunskill (2016), Swaminathan & Joachims (2015)) however, since control variates are generally considered to preserve (asymptotic) qualities of estimators, not all OPE estimators proposed for ORL can be defined, strictly, as control variate methods i.e., not all of the aforementioned estimators preserve such behaviours.

Equation 1 defines an empirical approximation to the generic RL objective (equation 13 in Uehara et al. (2022)), however, it is expressive enough to capture the various OPE estimators which exist in

the literature.

$$J_{\tau \sim \pi_e, \mathcal{M}}(d) = g_2(\cdot) \sum_{i=1}^{n} \left( \sum_{t=0}^{H_i - 1} \left( \gamma^t r(a_{i,t}, s_{i,t}) g_1(f(\{\pi_e(a_{i,t}|s_{i,t})\}_{0:H_i-1}, \{\hat{\pi}_\beta, (a_{i,t}|s_{i,t})\}_{0:H_i-1}, \cdot), \cdot)) \right) \right) \tag{1}$$

where $d$ defines an offline dataset generated under $\pi_\beta$ and $\mathcal{M}$ and $n = |d|$. $g_1(\cdot)$ defines the normalisation constant for the trajectory level importance samples and $g_2(\cdot)$ defines the normalisation constant for the empirical average[1]. Equations 2, 3, 4 and 5 define the hierarchy of steps for any IS estimator:

$$\frac{\pi_e(a_t|s_t)}{\pi_\beta(a_t|s_t)} : \forall t \in 0, ..., H - 1, \forall \tau \in d \tag{2}$$

$$f(\{\pi_e(a_t|s_t)\}_{1:H}, \{\hat{\pi}_\beta, (a_t|s_t)\}_{1:H}, \cdot) : \forall \tau \in d \tag{3}$$

$$g_1(f, \cdot) : \forall \tau \in d \tag{4}$$

$$g_2(\cdot) : \forall \tau \in d \tag{5}$$

Equation 2 is the same for all IS estimators currently implemented within offline_rl_ope however, equation 2 could be altered for state importance sampling methods. Equation 3 can be altered to define the per-decision IS estimator (Precup et al. (2000)). Equations 4 and 5 are used to define the various approaches to weighted IS based estimators (including DR estimators). A full breakdown of common IS estimators is defined in table 2. Let $\rho_{\mathrm{IS},i,t}$ and $\rho_{\mathrm{PD},i,t}$ define the vanilla IS and per-decision importance samples for trajectory $i$ and timestep $t$, respectively:

$$\rho_{\mathrm{IS},i,t} = \prod_{t=0}^{H_i} \frac{\pi_e(a_{i,t}|s_{i,t})}{\pi_\beta(a_{i,t}|s_{i,t})} : \forall t \in 0, .., H - 1, \forall \tau \in d$$

$$\rho_{\mathrm{PD},i,t} = \prod_{t'=0}^{t} \frac{\pi_e(a_{i,t'}|s_{i,t'})}{\pi_\beta(a_{i,t'}|s_{i,t'})} : \forall t \in 0, .., H - 1, \forall \tau \in d$$

Note that, for a fixed $i$, $\rho_{\mathrm{IS},i,t}$ is constant $\forall t \in 0, ..., H_i - 1$. Herein $\rho_{X,i,t} = \rho_{\mathrm{IS},i,t}$ or $\rho_{X,i,t} = \rho_{\mathrm{PD},i,t}$ depending on the context. Additionally $n$ defines the total number of trajectories and $H_i$ defines the length of trajectory $i$.

Additionally to those defined in table 2, it is common practice to 'clip' importance weights which could conceivably be implemented at any stage of the aforementioned hierarchy. Clipping in offline_rl_ope is performed in between equations 3 and 4 and is defined as:

$$\min(\max(w_f, w_{\mathrm{clip}}^{-1}), w_{\mathrm{clip}}) : \forall w_f \tag{6}$$

where $w_f \in \{f(\{\pi_e(a_t|s_t)\}_{1:H}, \{\hat{\pi}_\beta, (a_t|s_t)\}_{1:H}, \cdot) : \forall \tau \in d\}$ and $w_{\mathrm{clip}}$ is defined a priori.

Finally, to ensure stability of the weighted importance sampling, offline_rl_ope integrates Laplacian smoothing. Smoothing can be included in any weighted calculation, for both equations 4 and 5 and is applied as the final stage of defining the denominator in all cases. For example, when applied to self-normalised weights in equation 5, the calculation would be:

$$\left( \epsilon + \sum_{i=1}^{n} \rho_{X,i,H} \right)^{-1}$$

Figure 1 depicts how the various elements of a standard OPE pipeline are implemented in offline_rl_ope. Currently, the only direct method implemented is FQE, which utilises the d3rlpy integration. As such, the proceeding primarily discusses the API with respect to IS and DR estimators.

---

[1]"·" here refers to arbitrary parameters defined later

Table 2: Mapping of estimator definitions to Equation 1 and literature references

| Name | Equ. | Implementation | Reference |
|---|---|---|---|
| Vanilla one step | Equ. 3 | $\rho_{\text{IS},i,t}$ : 
 $\forall t \in 0, .., H, \forall i \in 1, ..., n$ | Precup et al. (2000) 
 Kallus & Uehara (2019) 
 Jiang & Li (2016) |
| Per-decision | Equ. 3 | $\rho_{\text{PD},i,t}$ : 
 $\forall t \in 0, .., H, \forall i \in 1, ..., n$ | Precup et al. (2000) 
 Kallus & Uehara (2019) |
| Identity | Equ. 4 | $\rho_{X,i,t} : \forall t \in 0, .., H, \forall i \in 1, ..., n$ | Precup et al. (2000) |
| Vanilla norm of Equ. 4 | Equ. 4 | $n^{-1} : \forall t \in 0, .., H, \forall i \in 1, ..., n$ | Thomas & Brunskill (2016) |
| Point in Time self-normalised | Equ. 4 | $(n_t)^{-1} \sum_{i=1}^{n_t} \mathbf{1}_{p_{i,t}>0}(\rho_{X,i,t})\rho_{X,i,t}$ : 
 $\forall t \in 0, .., H, \forall i \in 1, ..., n$ | Kallus & Uehara (2019) 
 Thomas & Brunskill (2016) |
| Vanilla norm of Equ. 5 | Equ. 5 | $n^{-1} : \forall t \in 0, .., H, \forall i \in 1, ..., n$ | Precup et al. (2000) 
 Kallus & Uehara (2019) 
 Jiang & Li (2016) |
| Self-normalised | Equ. 5 | $\left( \sum_{i=1}^{n} \rho_{X,i,H} \right)^{-1}$ : 
 $\forall i \in 1, ..., n$ | Precup et al. (2000) |
| Cumulative (discount) self-normalised | Equ. 5 | $\left( \sum_{i=1}^{n} \sum_{t=0}^{H-1} \rho_{X,i,t} \right)^{-1}$ : 
 $\forall t \in 0, .., H, \forall i \in 1, ..., n$ | Precup et al. (2000) |

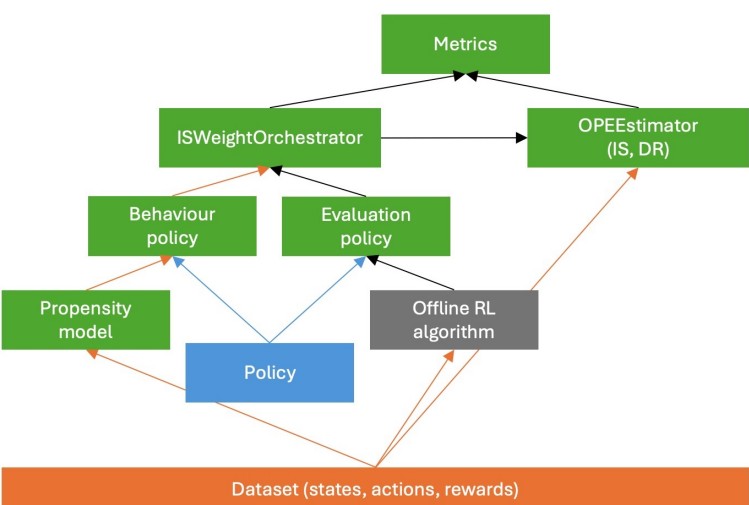

Figure 1: Flowchart of components for performing OPE with IS based estimators in offline_rl_ope. Key: green squares define classes within offline_rl_ope that perform a calculation; blue squares define helper classes within offline_rl_ope; orange squares defined fixed external inputs; grey squares define changing external inputs; black arrows defines changing information; orange arrows define fixed information (conditional on the fixed input); blue arrows define helper funtionality relationships, not information flow.

## 3.1 ESTIMATION API

The estimation API defines the calculation mechanics of all the estimators described in section 3 as well as DR, DM and any additional methods added in future releases. This hierarchy described in section 3 is utilised to: improve computation time when using multiple IS estimators (since reused outputs can be cached); to enable custom estimators to be implemented with minimal additional code; and to streamline testing and code maintenance. The core elements of the API are described below and notable attributes/methods are described in table 3.

**ISWeightCalculator**  A single ISWeightCalculator object is defined per behaviour policy. The ISWeightCalculator class handles querying the evaluation and behaviour policy; calculating the one-step importance ratios (equation 2); and caching of weights to be used across multiple estimators, reducing computation. Additionally, the ISWeightCalculator automatically defines and caches the lengths of each trajectory, ensuring datasets with uneven trajectory lengths can be used without preprocessing from the user.

**ImportanceSampler**  Child classes of ImportanceSampler implement equation 3 e.g., the VanillaIS class defines Vanilla one step importance sampling whilst PerDecisionIS implements the Per-decision estimator. When using multiple different ImportanceSampler objects for a single behaviour policy (e.g., when performing vanilla IS and per-decision importance sampling) the ISWeightOrchestrator (which is a child class of ISWeightCalculator) can be used to facilitate the sharing of one-step weights across multiple instances of ImportanceSampler. This ensures the behaviour and evaluation policies are only queried once, thus reducing computation.

**ISEstimatorBase/WeightDenomBase**  The ISEstimatorBase class implements the mechanics of estimating the reward of a single trajectory whilst child classes, (e.g., ISEstimator and DREstimator) implement the specific calculation (as per equation 1). Critically, any ISEstimatorBase object requires WeightDenomBase for instantiation where child classes of WeightDenomBase implement equation 4.

**OPEEstimatorBase/EmpiricalMeanDenomBase**  The OPEEstimatorBase implements the mechanics of summarising the trajectory level rewards (defined by ISEstimatorBase) across an entire dataset. This broadly requires summing the trajectory level rewards and applying variations of equation 5, defined by child classes of EmpiricalMeanDenomBase, which are required to instantiate an OPEEstimatorBase object.

Table 3: Notable classes and associated methods and attributes.

| Class | Methods attribute | Description |
|---|---|---|
| ISWeightCalculator & ISWeightOrchestrator (Equation 2) | is_weights | Tensor of dimension $(n, \max[H_i])$ of one-step importance ratios |
| | weight_msk | Tensor of dimension $(n, \max[H_i])$ with value 0 after a trajectory has terminated |
| | update | Updates is_weights using the evaluation policy provided |
| ImportanceSampler (Equation 3) | traj_is_weights | Tensor of dimension $(n, \max H_\tau)$ of trajectory importance ratios. |
| | get_traj_weight_array | Abstract method requiring child classes to implement variations of equation 3 |
| ISEstimatorBase (Equation 4) | process_weights | Applies IS weight clipping (Equ. 6) and the calculation defined by WeightDenomBase |
| WeightDenomBase (Equation 4) | __call__ | Abstract method requiring child classes to implement variations of equation 4 |
| OPEEstimatorBase (Equation 5) | predict_traj_rewards | Abstract method requiring child classes to implement estimator mechanics i.e., doubly robust vs pure importance sampling |
| | predict | Core public method for calculating the dataset estimate |
| EmpiricalMeanDenomBase (Equation 5) | __call__ | Abstract method requiring child classes to implement variations of equation 5 |

## 3.2 POLICY

The BasePolicy class defines a standardised API for obtaining state-action probabilities under a given policy. Shipped with offline_rl_ope are the Policy and GreedyDeterministic classes which define framework agnostic wrappers for stochastic and greedy deterministic polices, respectively, for functions returning both Pytorch tensors and numpy arrays. The irl_example.py script in the code accompanying this paper provides an example of how the Stable Baselines3 (Raffin et al. (2021)) policy API can be made compatible with offline_rl_ope through a simple wrapper class. To enable ease of debugging and monitoring, the BasePolicy class optionally allows policy outputs to be easily cached, similarly to the ISWeightCalculator and ImportanceSampler APIs.

## 3.3 ADDITIONAL NOTABLE FUNCTIONALITY

### 3.3.1 PROPENSITY MODELS

The majority of recent OPE applications entail large state (and action) spaces and as such, require defining the behaviour policy via function approximation (Hanna et al. (2019)). offline_rl_ope provides an API for defining propensity models with Pytorch (Paszke et al. (2019)) and scikit-learn (Pedregosa et al. (2011)).

### 3.3.2 APIs

**3rd Party Integration** Whilst the focus of offline_rl_ope was on defining a standalone OPE framework, providing optional integrations with popular ORL workflows was deemed a necessity. Currently offline_rl_ope is (optionally) tightly integrated with d3rlpy (Seno & Imai (2021)). The existing implementation allows any OPE estimator defined with offline_rl_ope to be used to assess d3rlpy models both post and during training. In particular the "during training" API aligns with the recommendations of Tang & Wiens (2021) as it enables an early stopping type workflow. Running this workflow is further aided by the caching of reusable computations discussed in section 3.1.

**Plug and play** offline_rl_ope has been designed to be trivally extendable by defining low level modules for constructing OPE estimators (section 3.1). However, in order to address the consistency issues described in section 1, a plug and play API has been additionally provided.

### 3.3.3 OFFLINE OPE METRICS

Effective sample size (ESS) is a metric colloquially associated with IS methods with the intent of describing the "(potentially) reduced information content of a dataset given an evaluation policy". For example, Liu et al. (2022) utilised the ESS definition in equation 7, from Owen (2013).

$$\text{ESS} = \frac{n}{1 + \text{cv}(w)^2} \tag{7}$$

Such that:

$$w_i = \frac{\pi_e(a_i|s_i)}{\pi_\beta(a_i|s_i)}; \text{cv}(w) = \frac{\text{sd}_w}{\bar{w}}; \bar{w} = \frac{1}{n}\sum_{i=1}^{n} w_i; \text{sd}_w = \sqrt{\left(\frac{1}{n-1}\sum_{i=1}^{n}(w_i - \bar{w})^2\right)}$$

Such a definition, along with others (such as that proposed by Kong (1992)) have been designed for performing Monte Carlo Integration in a fundamentally different contexts to OPE. The diagnostics for non OPE Monte Carlo IS have been derived under the assumption that the importance distribution ($p_{\pi_\beta}$ in OPE) is variable and the nominal distribution ($p_{\pi_e}$ in OPE) is fixed. Owen (2013) derived diagnostics by utilising the fact that the variance of a Monte Carlo IS estimator can be defined as:

$$\text{Var}[J_{\text{IS}}(\pi_e; \tau)] = \int_{\{\tau:p_{\pi_\beta}(\tau)>0\}} \frac{\left(p_{\pi_e}(\tau)\sum_{t=0}^{\infty} r_t\gamma^t - \mu p_{\pi_\beta}(\tau)\right)^2}{p_{\pi_\beta}(\tau)} d\tau \tag{8}$$

where $\mu = \mathbb{E}_\tau[J_{IS}(\pi_e; \tau)]$. However, focused on monitoring the $p_{\pi_\beta}$ terms, since these could have been altered to reduce the variance. For OPE however, the behaviour policy is fixed and thus in order to reduce the variance and "obtain a higher effective sample size", the deviations between the behaviour and evaluation policy should be reduced, as described by the $\left(p_{\pi_e}(\tau) \sum_{t=0}^{\infty} r_t \gamma^t - \mu p_{\pi_\beta}(\tau)\right)^2$ term. Appendix D demonstrates how the diagnostics used to monitor the importance distribution (such as equation 7) produce undesirable results for OPE.

**VWP** Motivated by monitoring the *symmetric deviations* between the importance and nominal distribution, the metric "VWP" (valid weight proportion) is proposed. Utilising the fact that $\sum_{t=0}^{\infty} \frac{p_{\pi_e}(s_t, a_t)}{p_{\pi_\beta}(s_t, a_t)} \propto (p_{\pi_e}(\tau) \sum_{t=0}^{\infty} r_t \gamma^t - \mu p_{\pi_\beta}(\tau))^2$, let:

$$\text{VWP} = \frac{1}{n} \sum_{i=1}^{n} \mathbf{1}_{w_{\min} \leq w_i \leq w_{\max}}(w_i) \tag{9}$$

where $w_i = \sum t = 0^\infty \rho_{IS,i,t}$ or $w_i = \sum_{t=0}^{\infty} \rho_{PD,i,t}$ depending on the context and the desirable behaviour is for VWP $\rightarrow 1$ as $w_{\min} \rightarrow 0$ and $w_{\max} \rightarrow \infty$. VWP ignores the dependence on $\mu$ in equation 8 however, the metric does overcome the described failure modes of ESS.

**WeightStd** In addition to using VWP, a metric for tracking the standard deviation of weights (WeightStd) is also implemented within offline_rl_ope. WeightStd is defined as per $sd_w$, above, i.e.:

$$\text{WeightStd} = \sqrt{\left(\frac{1}{n-1} \sum_{i=1}^{n} (w_i - \bar{w})^2\right)} \tag{10}$$

Both VWP and WeightStd measure the deviation of weights however, in contrast to VWP, WeightStd centers around the mean deviation rather than 1. A mean deviation of 1 is significant as it represents the minimal deviation from the behaviour policy and thus minimal additional generalisation error. Whilst a standard deviation of 1 would also represent such a scenario, WeightStd is unable to distinguish between a policy that systematically deviates from the behaviour policy at a constant magnitude; and a policy which remains close to the behaviour policy but deviates significantly at a small subset of trajectories. This is a result of the relatively larger impact that outliers can have on the mean calculation. However, when used in conjunction with VWP, the WeightStd can identify such scenarios since the former would present with a slow VWP convergence whilst the latter would present with a faster rate of convergence. Further, distinguishing between these scenarios is important as uncertainty in the latter policy can be reduced using weight clipping without greatly affecting the overall behaviour of the policy. Table 4 provides an overview of how VWP and WeighStd can be jointly interpreted to debug IS weights.

Table 4: Joint interpretation of VWP and WeightStd metrics

| Scenario | VWP | WeightStd | Interpretation |
|----------|-----|-----------|----------------|
| 1 | $\rightarrow 0$ | $\rightarrow 0$ | Uncertainty due to consistent divergence from the behaviour policy. Constrain entire policy to reduce uncertainty. |
| 2 | $\rightarrow 0$ | $\rightarrow \infty$ | Maximal uncertainty due to consistent divergence from the behaviour policy and the estimation is dominated by a subset of trajectories. Constrain entire policy to reduce uncertainty. |
| 3 | $\rightarrow 1$ | $\rightarrow 0$ | Minimal uncertainty |
| 4 | $\rightarrow 1$ | $\rightarrow \infty$ | Uncertainty due to estimate being dominated by a subset of trajectories. Implement weight clipping at reasonable order of magnitude from 1. |

## 4    ACCURACY OF IMPLEMENTATION

All estimators implemented within offline_rl_ope have been unit tested however, additional analysis was conducted (where possible) to ensure consistency of implementation.

### 4.1    DISCRETE ACTION ESTIMATORS

The implementations of discrete action (continuous state) estimators were compared across offline_rl_ope and Scope-RL. Table 5 demonstrates that the implementations for: IS, WIS, PD WPD, DR and WDR estimators did not differ materially.

Table 5: Comparison of offline_rl_ope and Scope RL estimations in a continuous state-discrete action environment, RTBGym (Kiyohara et al. (2023))

| Estimator | Mean difference (Scope-RL denom) | Mean difference (OPO denom) |
|---|---|---|
| IS | 0.00% | 0.00% |
| WIS | 0.00% | 0.00% |
| PD | 0.00% | 0.00% |
| WPD | 0.00% | 0.00% |
| DR | 0.00014% | 0.00014% |
| WDR | 0.0028% | 0.0028% |

### 4.2    CONTINUOUS ACTION SPACES

With respect to continuous action spaces, offline_rl_ope and Scope-RL differed significantly in their approach and as such, could not be compared against one another. To demonstrate the efficacy of the offline_rl_ope implementation for continuous actions spaces, the relative ranking[2] of 3 policies were compared against the ground truth evaluations using the Pendulum environment (Towers et al. (2024)). In addition to a number of other expected observations, table 6 suggests that broadly speaking, estimators implemented in offline_rl_ope were able to accurately rank the performance of policies against the ground truth performance, demonstrating the efficacy of implementation. In addition, expected observations included:

- Pure PD estimators benefited the most from weight clipping since the bias of doubly robust methods was already being controlled through the reward approximation;
- The pure PD estimator demonstrated the worse correlation due to the high variance of the estimator;
- Combining the FQE method from d3rlpy with the DR and WDR methods improved the ranking performance, despite all FQE models converging reasonably well and a reasonable amount of hyperparameter tuning being performed (figures 3 in appendix E).

To conclude, despite the lack of existing benchmark for performing OPE on continuous stochastic policies, the results and observations highlighted (in addition to the unit testing performed) provided reasonable evidence as to the efficacy of implementation within offline_rl_ope.

## 5    EXAMPLE USE OF VWP AND WEIGHTSTD METRICS (CONTINUOUS ACTION SPACE)

To demonstrate the efficacy of the VWP and WeightStd metrics, these were used to integrogate the ranking performance of OPE estimators utilising, per-decision weights, over policies from the continuous action task described in section 4.2. Table 7 compares the ranking performance of using non-clipped estimators against an average of 6 clipped estimators (at different magnitudes). Overall,

---

[2]Since OPE estimator prediction is heavily dependant on the problem context, policy ranking was deemed sufficient to demonstrate the implementation efficacy.

Table 6: Average (across 5 random seeds) correlation of policy rankings in comparison to environment ground truth. OoM describes the order of magnitude of clipping applied.

| | Spearman Correlation | | |
|---|---|---|---|
| Estimator | No clipping | Clipping OoM 1 | Clipping OoM 2 |
| IS | Undefined | Undefined | Undefined |
| WIS | Undefined | Undefined | Undefined |
| PD | -0.1 | 0.4 | 0.3 |
| WPD | 0.5 | 0.8 | 0.7 |
| DR | 0.7 | 0.7 | 0.7 |
| WDR | 0.7 | 0.6 | 0.6 |
| DM | 0.5 | NA | NA |

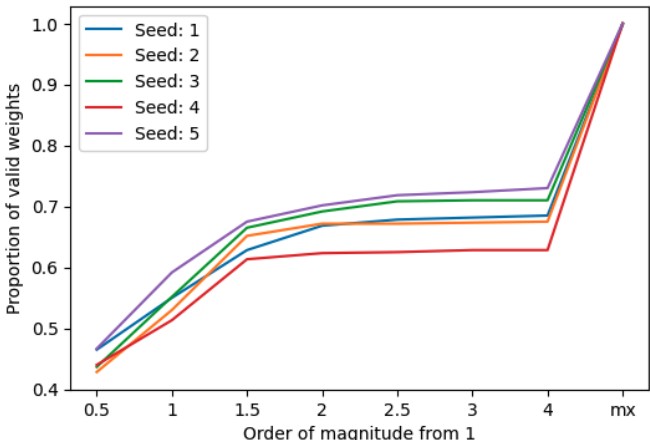

Figure 2: Caption

it was clear that the sampling uncertainty of the underlying dataset effected the performance of the OPE estimator, most notably in seed 2 (even after weight clipping) and in seed 5 where weight clipping significantly boosted performance.

Table 7: Average (over policies) rank performance of policies against ground truth performance

| Seed | Spearman's R (no clipping) | Spearman's R (with clipping) |
|---|---|---|
| 1 | 0.88 | 0.88 |
| **2** | **-0.50** | **-0.29** |
| 3 | 0.50 | 0.62 |
| 4 | 1.00 | 0.92 |
| **5** | **0.38** | **0.71** |

Figure 2 displays the VWP metrics, averaged across policies for each seed. Notably, seed 5 had a relatively quicker VWP convergence rate whilst seed 2 had a relatively low one. Additionally, the mean WeightStd values (across policies) for seed 2 and seed 5 were 10.21 and 215.80, respectively (full figures in table 9). Arguably, the policies within seed 2 aligned to scenario 1 in table 4, where the poor predictions were a result of the policies systematically diverging from the behaviour policy. In contrast, the policies in seed 5 aligned to scenario 4, where the poor predictions were a result of the uncertainty induced by a subset of trajectories. As such, the uncertainty in OPE estimates for seed 5 had the potential to be reduced through weight clipping to a greater extent than for seed 2.

Utilising VWP and WeightStd to interogate the performance of OPE estimates is a probabilistic exercise. According to figure 2, the policies in seed 4 were divergent (due to the low VWP) however,

the OPE rankings were very accurate. This was most likely a result of the good performance of the OPE estimators when clipping was not applied. Table 8 describes the results of a logistic regression model, assessing the relationship between:

- The amount of clipping and;

- The original performance of the unclipped estimator;

against the probability that an additional order of magnitude of weight clipping harmed the ranking performance of the estimator. Notably, the higher the ranking performance of the original estimator, the less likely that clipping was to harm the performance, providing evidence for the hypothesis regarding the policies in seed 4.

Table 8: Coefficients and p-values (t-test), measuring the linear relationship between the magnitude of weight clipping applied to an estimator plus the original ranking performance of the un-clipping estimator agains the probability that the current magnitude of clipping harms the ranking performance

| Name | Coefficient | P-value |
|---|---|---|
| Intercept | 0.11 | 0.86 |
| Amount of clipping | -0.95 | 0.34 |
| Performance of unclipped estimator | -2.11 | 0.00 |

Similar logistic regression tests were performed on combinations of WeightStd and VWP at different orders of magnitude (table 10 in appendix E). Whilst the results were encouraging with respect to the direction of the coefficients, the significance of the effect sizes were inconclusive, suggesting further work is required to understand the true predictive nature of the metrics.

## 6 NEXT STEPS

Next steps for the development of offline_rl_ope would be to implement additional OPE estimation techniques, develop additional non-oracle metrics for assessing OPE estimations as well as further assessing the predictive power of VWP and WeightStd. An interesting area of future research for IS estimators with regression models would be to develop uncertainty estimates which combine both the uncertainty in estimation of the propensity model and the resulting OPE estimation. A limitation of the existing offline_rl_ope implementation is the over-reliance on PyTorch. Whilst this has simplified integration with other PyTorch frameworks, the implementation restricts integration with other popular frameworks such as Tensorflow and Jax. Additionally, performance improvements in the computation of IS estimates from multi-processing could be explored.

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

## A  OFFLINE_RL_OPE/SCOPE-RL DISCUSSION

Table 1 provides an overview of the differentiating factors between offline_rl_ope and Scope-RL however, these are discussed in greater detail below. The primary difference is "audience" which is a result of a number of features being included in offline_rl_ope, not included in Scope-RL.

### A.1  AUDIENCE

**Uneven trajectory lengths**  The Scope-RL API requires trajectories to be a constant length, an assumption which is certainly not always satisfied in real world applications. Whilst trajectories could be padded prior to running OPE, this would render the calculation of self-normalised estimates incorrect. In comparison, offline_rl_ope can handle trajectories of differing lengths and does not require the user to perform any pre-processing in order to do so (see section 3.1 for a further explanation).

**Generic workflow specifically for real world data**  In addition to supporting uneven trajectory lengths, the offline_rl_ope API enables real world applications in several other ways. offline_rl_ope supports arbitrary evaluation and behaviour policies through the Policy class (section 3.2) whilst providing functionality to trivially define Pytorch and sklearn behaviour estimators through the PropensityModels API (section 3.3.1). In contrast, Scope-RL is deeply integrated with d3rlpy and does not provide any functionality for defining behaviour propensity estimates outside of the generation of synthetic datasets. Additionally, the OPE evaluation metrics included in Scope-RL require an oracle measure of performance. In contrast, offline_rl_ope focuses on defining a workflow for real world data by introducing metrics (section 3.3.3) such as VWP and enabling stages of the OPE pipeline to be easily cached for debugging and post-hoc analysis.

**Off-policy selection with d3rlpy**  Despite having a generic API, offline_rl_ope ships, natively, with a deep integration with d3rlpy. In particular, offline_rl_ope enables OPE metrics to be run during training (see section 3.3.2) thus enabling early stopping to be performed.

### A.2  ESTIMATORS, API AND IMPLEMENTATION

In addition to audience, Scope-RL differs in the number of other areas. The number of estimators currently implemented within Scope-RL far exceeds that of offline_rl_ope in particular, state marginal IS estimators, double reinforcement learning and the DICE family of estimators. Whilst these estimators will be implemented within offline_rl_ope in the future, users looking for implementations as of the time of writing this document are referred to Scope-RL.

The offline_rl_ope API utilises equation 1 to enable an array of IS based estimators to be implemented, relying on significant class inheritance. This has resulted in an API which is easy to extend and maintain and is in contrast to the individually defined discrete and continuous estimators within the Scope-RL API. The implementation deatils of this API are discussed in section 3.1.

The final difference between Scope-RL and offline_rl_ope is the handling of continuous action spaces. offline_rl_ope can only (reasonably) evaluate stochastic policies learnt over a continuous action spaces whilst, in Scope-RL, all policies over continuous actions spaces are evaluated through kernel smoothing of actions (Kallus & Zhou (2018)).

## B    MOTIVATING THE USE OF IS ESTIMATORS

The objective of OPE is to evaluate the expected discounted reward of a policy $\pi_e$, $\mathbb{E}_{\tau \sim p_{\pi_e}}[\sum_{i=0}^{\infty} r_i \gamma^i]$. Pure IS estimators refer to any OPE estimator defined as per equation 1. DR estimators considered in offline_rl_ope can also be defined similarly to equation 1 however, for an estimator to be doubly robust, certain bias conditions must also be met. Generally speaking DR estimators incorporate a (direct) approximation of the policy value under $\pi_e$ i.e., $\hat{Q}_{\pi_e}(s, a)$ in addition to the IS estimates. In contrast, direct methods (DM) solely utilise the aforementioned value function estimate.

Within ORL, the FQE DM method (Le et al. (2019)) is often cited as a "go-to" method for performing OPE (Voloshin et al. (2021)). However, theoretical and empirical evidence suggests that the selection of OPE estimator is problem specific. Theoretically, the decision of whether to use an IS estimator or a direct method estimator is a function of the complexity of the propensity and reward (outcome) model, respectively (Alaa & van der Schaar (2018)). Voloshin et al. (2021) observed this empirically as well as noting additional factors such as: evaluation policy/behaviour policy mass-match and horizon length.

There also exists practical differences in estimators, in terms of complexity of hyperparameter tuning and computation time. For pure IS methods, the aforementioned complexities are a result of the behaviour policy estimation and as such, need to only be performed once per behaviour policy (rather than per evaluation policy as in the case of direct and DR methods), pure IS methods are more suited for rapid model experimentation. Tang & Wiens (2021) leveraged this observation, proposing a two stage model development pipeline, where IS methods are used for initial model assessment.

The above observations clearly motivate the development of a robust code base for performing a range of OPE estimation.

## C    DEFINING IS/DR ESTIMATORS IN OFFLINE_RL_OPE

Using the definitions provided in section 3, sudo code for defining different IS estimators is provided below. The vanilla IS estimator has been defined using the low level API whilst the WIS and WDR estimators have been defined using the plug and play API. The "rewards", "states" and "actions" parameters except lists of PyTorch Tensors, the discount parameter excepts a float value and the behav_policy and eval_policy except classes of type offline_rl_ope.components.Policy.Policy.

```
from offline_rl_ope.components import ISWeightOrchestrator
from offline_rl_ope.OPEEstimators import ISEstimator
from offline_rl_ope.OPEEstimators import (
    ISEstimator, EmpiricalMeanDenom,
    PassWeightDenom, WeightedEmpiricalMeanDenom
    )
from offline_rl_ope.api.StandardEstimators import (
    VanillaISPDIS, WIS, WDR)

vanilla_est = ISEstimator(
    empirical_denom=WeightedEmpiricalMeanDenom(),
```

```
702         weight_denom=PassWeightDenom ( )
703         )
704
705     w_est = WIS()
706     smthd_w_est = WIS( smooth_eps =0.0000001)
707
708     w_dr_est = WDR(
709         dm_model =.
710     )
711
712     is_calc = ISWeightOrchestrator (
713         "vanilla",
714         "per_decision"
715         behav_policy =.
716         )
717
718     is_calc . update (
719         states =.,
720         actions =.,
721         eval_policy =.
722     )
723
724     vanilla_is = vanilla_est . predict (
725         rewards =.,
726         discount =.,
727         weights=is_calc [" vanilla "]. traj_is_weights ,
728         is_msk=is_calc . weight_msk ,
729         states =.,
730         actions =.,
731     )
732
733     vanilla_pd = vanilla_est . predict (
734         rewards =.,
735         discount =.,
736         weights=is_calc [" per_decision "]. traj_is_weights ,
737         is_msk=is_calc . weight_msk ,
738         states =.,
739         actions =.,
740     )
741
742     vanilla_wis = w_est . predict (
743         rewards =.,
744         discount =.,
745         weights=is_calc [" per_decision "]. traj_is_weights ,
746         is_msk=is_calc . weight_msk ,
747         states =.,
748         actions =.,
749     )
750
751     smoothed_wis = smthd_w_est . predict (
752         rewards =.,
753         discount =.,
754         weights=is_calc [" per_decision "]. traj_is_weights ,
755         is_msk=is_calc . weight_msk ,
        states =.,
        actions =.,
    )

    w_dr = w_dr_est . predict (
```

```
        rewards =.,
        discount =.,
        weights=is_calc["vanilla"].traj_is_weights,
        is_msk=is_calc.weight_msk,
        states =.,
        actions =.,
)
```

## D  FAILURE MODES OF IS METRICS FOCUSED ON THE IMPORTANCE DISTRIBUTION

For simplicity, consider two evaluation policies $\pi_{e_1}$ and $\pi_{e_2}$. Let $w_1 = \{w_{1,i} = c_+ : i \mod 2 = 1 \forall i \in 1, ..., n\} \cup \{w_{1,i} = c_{++} : i \mod 2 = 0 \forall i \in 1, ..., n\}$ define the set of importance sample weights for $n$ trajectories associated with evaluation policy $\pi_{e_1}$. Let $w_2 = \{w_{1,i} = c_+ : i \mod 2 = 1 \forall i \in 1, ..., n\} \cup \{w_{1,i} = c'_+ : i \mod 2 = 0 \forall i \in 1, ..., n\}$ define the set of importance sample weights for $n$ trajectories associated with evaluation policy $\pi_{e_2}$. Additionally let $c_{++} = c_+ + \epsilon$ and $c_+ = (c'_+ + \epsilon)^{-1}$.

In words, policy $\pi_{e_1}$ and $\pi_{e_2}$ deviate to equal extents from $\pi_\beta$, the difference being $\pi_{e_2}$ is symmetric. Let ESS be defined as per equation 7 then the metric is defined by the value of $\text{cv}(w)^2$. For $\pi_{e_1}$ and $\pi_{e_2}$ this equals:

$$\text{cv}(w_1)^2 = \left(\frac{\sqrt{\frac{n}{4n-1}\epsilon^2}}{c_+ + \frac{1}{2}\epsilon}\right)^2$$

$$\text{cv}(w_2)^2 = \left(\frac{\sqrt{\frac{n}{n-1}(\frac{1}{2}c_+ + \frac{1}{n\epsilon})^2}}{2(n\epsilon)^{-1}}\right)^2$$

And therefore, as $c_+ \to \infty$, $\text{cv}(w_1)^2 \to 0$ and $\text{cv}(w_2)^2 \to \infty$. Following from this, as $c_+ \to \infty$, $\text{ESS}(w_1) \to m$ whilst $\text{ESS}(w_2) \to 0$. However, regardless of the value of $c_+$, both policies $\pi_{e_1}$ and $\pi_{e_2}$ should be defined equally in terms of the "(potentially) reduced information content of a dataset given an evaluation policy".

## E  SUPPORTING FIGURES FOR CONTINUOUS CONTROL EXPERIMENT

The following section contains a number of supporting figures for the experiment discussed in sections 4.2 and **??**.

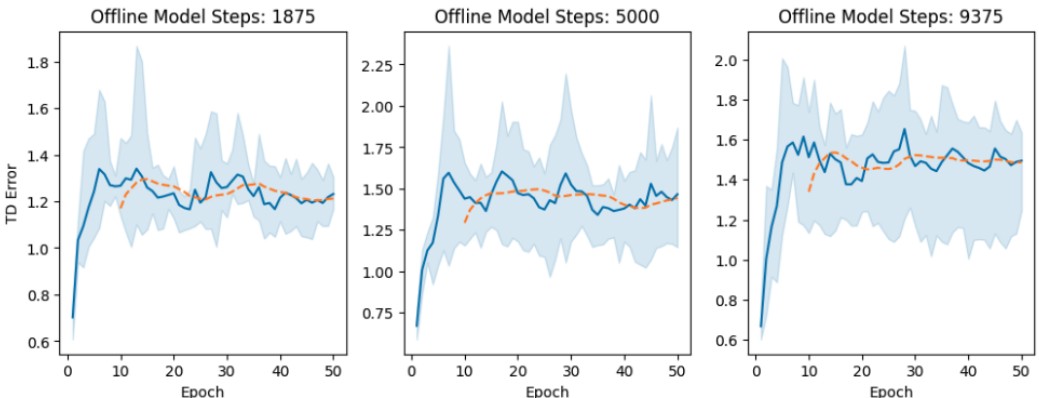

Figure 3: TD error for FQE (DM) models for each of the policies trained. The error bars are defined by the min and max TD error. The orange dotted line defines the 10 step moving average.

Table 9: WeightStd and VWP metric values for all policies (1875,5000 and 9375) across all seeds

| Seed | Policy | WeightStd | VWP 1 | VWP 2 | VWP 3 | VWP 4 | VWP 5 | VWP Max |
|------|--------|-----------|-------|-------|-------|-------|-------|---------|
| 1 | 1875 | 85.36 | 0.54 | 0.54 | 0.67 | 0.68 | 0.69 | 1 |
| 1 | 5000 | 49.73 | 0.55 | 0.55 | 0.68 | 0.69 | 0.69 | 1 |
| 1 | 9375 | 21.3 | 0.57 | 0.57 | 0.67 | 0.68 | 0.69 | 1 |
| 2 | 1875 | 10.92 | 0.54 | 0.54 | 0.68 | 0.68 | 0.68 | 1 |
| 2 | 5000 | 10.28 | 0.53 | 0.53 | 0.68 | 0.68 | 0.68 | 1 |
| 2 | 9375 | 9.43 | 0.52 | 0.52 | 0.67 | 0.67 | 0.68 | 1 |
| 3 | 1875 | 20.22 | 0.54 | 0.54 | 0.69 | 0.71 | 0.71 | 1 |
| 3 | 5000 | 29.43 | 0.54 | 0.54 | 0.69 | 0.71 | 0.71 | 1 |
| 3 | 9375 | 53.71 | 0.57 | 0.57 | 0.69 | 0.71 | 0.71 | 1 |
| 4 | 1875 | 776.37 | 0.5 | 0.5 | 0.62 | 0.62 | 0.62 | 1 |
| 4 | 5000 | 62.48 | 0.5 | 0.5 | 0.62 | 0.63 | 0.63 | 1 |
| 4 | 9375 | 11.46 | 0.55 | 0.55 | 0.62 | 0.63 | 0.63 | 1 |
| 5 | 1875 | 104.99 | 0.6 | 0.6 | 0.69 | 0.73 | 0.73 | 1 |
| 5 | 5000 | 105.6 | 0.6 | 0.6 | 0.7 | 0.73 | 0.73 | 1 |
| 5 | 9375 | 436.81 | 0.58 | 0.58 | 0.71 | 0.72 | 0.73 | 1 |

Table 10: Coefficients and p-values (t-test), measuring the linear relationship between the magnitude of weight clipping applied to an estimator plus the VWP and WeightStd estimates against the probability that the current magnitude of clipping harms the ranking performance

| VWP Order of Magnitude | Name | Coefficient | P-value |
|---|---|---|---|
| 0.5 | Intercept | 41.77 | 0.02 |
| | Amount of clipping | -0.74 | 0.39 |
| | WeightStd | -0.00 | 0.16 |
| | VWP | -95.61 | 0.02 |
| 1 | Intercept | 52.04 | 0.00 |
| | Amount of clipping | -0.78 | 0.38 |
| | WeightStd | -0.02 | 0.00 |
| | VWP | -95.59 | 0.00 |
| 1.5 | Intercept | -15.11 | 0.31 |
| | Amount of clipping | -0.62 | 0.44 |
| | WeightStd | -0.00 | 0.46 |
| | VWP | 22.95 | 0.32 |
| 2 | Intercept | 31.65 | 0.14 |
| | Amount of clipping | -0.63 | 0.43 |
| | WeightStd | -0.02 | 0.02 |
| | VWP | -46.48 | 0.14 |
| 2.5 | Intercept | 27.64 | 0.05 |
| | Amount of clipping | -0.66 | 0.42 |
| | WeightStd | -0.02 | 0.00 |
| | VWP | -40.13 | 0.05 |
| 3 | Intercept | 33.40 | 0.02 |
| | Amount of clipping | -0.68 | 0.41 |
| | WeightStd | -0.02 | 0.00 |
| | VWP | -48.21 | 0.02 |

