# OpenReview forum: "offline_rl_ope: A Python package for off-policy evaluation of offline RL models with real world data"
_ICLR.cc/2025/Conference — Submitted to ICLR 2025_

### Official Review · Reviewer_46pa · 2024-11-01

**Soundness:** 1
**Presentation:** 1
**Contribution:** 1
**Rating:** 1
**Confidence:** 4

**Summary:**

The authors introduce a Python package for offline policy evaluation (OPE)
and discuss a number of improvements it offers such as handling uneven trajectory length, including novel metrics, providing effective API.
They also report experimental results for reproducibility check and some performance statistics.

**Strengths:**

It provides an extensive explanation of a new software package for OPE.

**Weaknesses:**

The paper is incomplete, poorly organized and inconclusive.
It is not ready for publication.

**Questions:**

NA

---

> ### Author Response · Authors · 2024-11-13
> **More specific feedback**
>
> Hi, I appreciate you reading the paper however, please could I request some more specific feedback regarding:
> - Why is it incomplete? What were you unconvinced by?
> - What conclusions did we leave open?
> - In what way was it poorly organised? Which parts did you not understand?
>
> Thanks

---

> > ### Comment · Reviewer_46pa · 2024-11-16
> >
> > * It is incomplete because no research question/problem is clearly raised or addressed. Also, the caption of Figure 2 is incomplete.
> > * There is no conclusion to be made since there is no research question/problem in the first place. Also, the experimental results are not enough to draw any clear conclusion as to whether `offline_rl_ope` is useful.
> > * The authors just put some new features and experimental results of the package in the paper without explicitly indicating any intention or explanation.
> >
> > Overall, it is more of a technical paper describing a software package, not a research paper.

---

> ### Author Response · Authors · 2024-11-21
> **Query responses**
>
> I think a core takeaway which I completely agree with is that the paper is not clear and we can definitely re-write to make clearer the issues you have highlighted with respect to research question, conclusion and results. With respect to your overall assessment however, I do agree that it is more of a technical paper however, the core contribution is the software package itself - there does not exist a well developed package for applying OPE to realworld data - this is research gap we are addressing.
>
> I do appriciate your comments though and will update the paper with them in mind.

---

> > ### Comment · Reviewer_46pa · 2024-11-22
> >
> > > the core contribution is the software package itself - there does not exist a well developed package for applying OPE to realworld data
> >
> > Thanks for the clarificaiton. I agree that this is a valuable and interesting research topic to pursuit.

---

### Official Review · Reviewer_hrrg · 2024-11-01

**Soundness:** 2
**Presentation:** 2
**Contribution:** 2
**Rating:** 3
**Confidence:** 3

**Summary:**

This paper describes a novel software library called offline_rl_ope to make real-world off-policy evaluation of RL policies easier. In particular, the python package (and paper) focus on Importance Sampling-based OPE methods (in contrast to fitted Q evaluation), as IS-based methods are missing a canonical implementation. The paper describes the problem of OPE, motivates the API design and functionality, and discusses common metrics for OPE.  Finally, the authors perform benchmarks against other existing software to ensure correctness.

Overall, I think the paper is interesting and potentially useful to the community. I have some questions about novelty, and what are the claimed contributions. Some of the results are difficult to evaluate. Perhaps the authors can help shed light on key questions that will better inform my decision.

I recommend rejecting the paper in its current form, as I do not believe it holds up to ICLR standard.

**Strengths:**

The paper proposes a software library with a good API. The abstractions utilized by the API are organized around important calculations in the OPE problem setup. The high-level code interface makes it easy for non-experts to evaluate their policies with simple python code.

The proposed library fills gaps in existing work (ie., Scope RL). In particular, facilitating evaluation of policies with uneven trajectories is a particularly useful feature.

The experimental cross-validation of the offline_rl_ope implementation and existing implementations is very strong and suggests the quality of the algorithms.

**Weaknesses:**

The overall contribution (while useful) seems small as other libraries do exist. Scope RL has a number of useful features, while d3rlpy has implemented FQE. I am not saying the proposed work has zero novelty; only that there is meaningful prior art in the space.




The claimed contributions are not entirely clear. I understand that the software package is new and how it compares to previous work. However, some of the contributions regarding metrics are not clear. For example, the VWP and WeightStd metrics are presented and experimentally validated. This would suggest that they are novel to the paper. Additionally, the authors state, “the metric ”VWP” (valid weight proportion) is proposed.” This leads me to believe that they are proposed here, but this verbiage is somewhat ambiguous as this also follows a discussion of ESS, which is not new. The major difficulty is that this contribution is stated neither in the abstract nor in the introduction, which casts doubt on the conclusion that they are novel to this paper.

The experimental validation of continuous action space in Section 4.2 is difficult to understand. The authors state, “offline rl ope and Scope-RL differed significantly in their approach and as such, could not be compared against one another.” Consequently, the authors only compare the relative ranking of the OPE outputs and compute the spearman correlation coefficient. While ranking is useful, I cannot assess the absolute quality of the OPE outputs under this condition, which casts doubt on the quality. Additionally, the authors state “estimators implemented in offline rl ope were able to accurately rank the performance of policies against the ground truth performance.” Some of the ranking statistics in Table 6 are fairly low (i.e., 0.3-0.5); I’m not sure the preceding statement is entirely true given this result.

The paper does not seem entirely complete. There are some typos and presentation issues. One of the most glaring problems is the empty caption in Figure 2. This error makes it seem like the paper was hastily written. Other typos:
- Line 427 “integrogate” should be interrogate
- Line 462: “effected” → “affected”

**Questions:**

What metrics are novel to this paper? Can the authors please state this clearly?
Why can they only compare the ranking of continuous actions? How do these implementations differ? Why do the spearman ranking correlation coefficients seem low in some instances?


Other suggestions:

The experiments in Section 5 reminded me of recent work on probabilistic policy ranking, which could potentially be incorporated into future releases:

Da, Longchao, et al. "Probabilistic Offline Policy Ranking with Approximate Bayesian Computation." Proceedings of the AAAI Conference on Artificial Intelligence. Vol. 38. No. 18. 2024.

---

> ### Author Response · Authors · 2024-11-15
> **Question responses**
>
> Thank you for your considered response and for taking the time to read our paper. We have responded to each of your questions below. We also completely take on board that the paper needs to be written more clearly.
>
> **What metrics are novel to this paper? Can the authors please state this clearly?**
>
> We would like to emphasise that the core contribution is:
> - A production ready Python package for performing offline OPE considering only assumptions that are valid in real world use cases;
> - The VWP and WeightStd form only one part of this contribution and were specifically designed since existing packages (i.e., ScopeRL) focus only on metrics requiring an oracle i.e., not "realworld".
>
> The metrics that are novel in this paper are valid weight proportion. In particular, the use of VWP in conjunction with an analysis of the standard deviation of weights to assess the validity of the IS estimates. The only metrics currently used to evaluate IS based OPE estimates for offline RL models are ESS. In particular, we are not proposing ESS here, in fact we propose it should not be used. As demonstrated by the example in the appendix of this paper (copied below), ESS is suboptimal for evaluating IS OPE estimators.
>
> Consider two evaluation policies $\pi_{e_{1}}$ and $\pi_{e_{2}}$. Let $w_{1} = \{w_{1,i} = c_{+}: i \mod 2 = 1 \forall i \in 1, ..., n\}\cup\{w_{1,i} = c_{++}: i \mod 2 = 0 \forall i \in 1, ..., n\}$ define the set of importance sample weights for $n$ trajectories associated with evaluation policy $\pi_{e_{1}}$. Let $w_{2} = \{w_{1,i} = c_{+}: i \mod 2 = 1 \forall i \in 1, ..., n\}\cup\{w_{1,i} = c_{+}': i \mod 2 = 0 \forall i \in 1, ..., n\}$ define the set of importance sample weights for $n$ trajectories associated with evaluation policy $\pi_{e_{2}}$. Additionally let $c_{++} = c_{+} + \epsilon$ and $c_{+} = (c_{+}' + \epsilon)^{-1}$.\\
>
> In words, policy $\pi_{e_{1}}$ and $\pi_{e_{2}}$ deviate to equal extents from $\pi_{\beta}$, the difference being $\pi_{e_{2}}$ is symmetric. Let $\textrm{ESS}$ be defined as per equation 7 then the metric is defined by the value of $\textrm{cv}(w)^2$. For $\pi_{e_{1}}$ and $\pi_{e_{2}}$ this equals:
> \begin{align*}
> \textrm{cv}(w_{1})^2 & = \Bigg(\frac{\sqrt{\frac{n}{4n-1}\epsilon^{2}}}{c_{+}+\frac{1}{2}\epsilon}\Bigg)^{2}\\
> \textrm{cv}(w_{2})^2 & = \Bigg(\frac{\sqrt{\frac{n}{n-1}(\frac{1}{2}c_{+}+\frac{1}{n\epsilon})^{2}}}{2(n\epsilon)^{-1}}\Bigg)^{2}
> \end{align*}
> And therefore, as $c_{+} \rightarrow \infty$, $\textrm{cv}(w_{1})^2 \rightarrow 0$ and $\textrm{cv}(w_{2})^2 \rightarrow \infty$. Following from this, as $c_{+} \rightarrow \infty$, $\textrm{ESS}(w_{1}) \rightarrow m$ whilst $\textrm{ESS}(w_{2}) \rightarrow 0$. However, regardless of the value of $c_{+}$, both policies $\pi_{e_{1}}$ and $\pi_{e_{2}}$ should be defined equally in terms of the "(potentially) reduced information content of a dataset given an evaluation policy".\\
>
> Since writing, it has come to our attention that variance metrics are used in contextual bandit IS estimators however, as demonstrated by  the results of section 5, solely relying on variance can give a false impression of performance when the evaluation policy concentrates far away from the behviour policy.
>
> **How do these implementations differ?**
> - offline_rl_ope can **only** handle stochastic continuous action spaces and compares the density functions of evaluation and behaviour policies (thereby preventing measure 0 evaluations i.e. $P(X=x)$ is measure 0 for continuous polices by p(x) is not where P is the cdf and p is the pdf);
> - In contrast, Scope RL assumes deterministic continuous action spaces and calculates IS estimates through kernel smoothing.
>
> It is the ambition that in the future, offline_rl_ope will include the option to perform kernel smoothing for deterministic continuous policies however, in the current release this is not available. That being said, the two methods give different policy evaluations and thus it was felt that comparing the two did not make sense since the aim of the comparison was to evaluate implementations.
>
> **Why can they only compare the ranking of continuous actions? Why do the spearman ranking correlation coefficients seem low in some instances?**
>
> Rankings were compared since OPE estimation is known to have high MSE see for example, Empirical Study of Off-Policy Policy Evaluation for Reinforcement Learning, Voloshin et al 2019 who also perform ranking/relative MSE. We would be more than happy to include the MSE results however, given the aim was to evaluate the implementations by observing expected trends (due to the lack of continuous baseline), ranking was chosen.
>
> **Relevance to Longchao, et al 2024**
> Thank you for pointing this reference - we do agree that the experiments are similar here. However, Longchao, et al 2024 seem to propose a novel method for off-policy selection where as the experiments presented in our paper are only to show the efficacy of implementations.

---

> > ### Comment · Reviewer_hrrg · 2024-11-26
> > **Response to the authors**
> >
> > I thank the authors for providing answers to my questions. In particular, I now have a better sense of 1) what metrics are novel to this paper, and 2) how offline_rl_ope differs from scope RL.
> >
> > Given that the paper likely needs a major revision, and in light of the scores of the other reviewers I maintain my score.

---

### Official Review · Reviewer_bhMC · 2024-11-03

**Soundness:** 2
**Presentation:** 1
**Contribution:** 2
**Rating:** 3
**Confidence:** 4

**Summary:**

The paper proposes a python package for off-policy evaluation methods. It has integrated common methods such as IS, WIS, PD, WPD, etc. The package supports multiple metrics and portable APIs for most of the classic methods. The paper also compares with a similar work **Scope-RL**, and shows the advantages of this work. Some details of the implementation of the existing work are discussed in the paper, which provides readers with necessary background information on the relevant techniques.

**Strengths:**

- The paper proposes a useful Python package for the off-policy evaluation methods in the RL domain, which can be helpful for an easy-to-use toolbox if one wants to implement an evaluation method quickly.
- The paper discussed some technique details of existing work, making readers out of the domain clearer on how the authors provide unified implementations.
- The author also provides a flowchart on how the package is designed and how each module is connected with each other.

**Weaknesses:**

- The paper is obviously written in a rush, with multiple unclear expressions and roughly created tables. For example, the caption text is not aligned between lines in Figure 1, the missing caption in Figure 2. Lack of explanations for equations  - see details in the questions section.
- Based on the content of the paper, it is not clear the significant contributions made by this work, while the unified framework for multiple off-policy evaluators is appreciated, it seems like the calibration of performance of the implemented methods is not mentioned, but it is crucial for a standard comparisons and potential users to care about.

> E.g1., authors can consider the use of Mean Absolute Error (MAE) to calculate the error between the OPE estimates and the ground truth for each method, while lower MAE indicates that the method is better calibrated.

> E.g2., another suggestion is the calibration curve: authors can consider generating the calibration visualization plots by showing the estimated returns (OPE results) against actual returns. In this test, a well-calibrated model should ideally fall on the diagonal line. This can provide better insights into how trustworthy this work's implementation is.
- The presentation in the paper is too simple and not informative.
E.g., it is unclear how many times the experiments have been done for the result report, from the table 6-7-8, are the values reported in the table average values or the experiment results from one execution? A more convincing way of conveying the results is by: mean ± std for a method's stable performance. Similarly for the content in the Figure 2.

**Questions:**

1. The format for the caption in Figure 1 seems not aligned, it is suggested to adjust for better presentation.
2. Figure 2 is not completed. The caption content is not described at all.
3. In line 794, what does `??` refers to?
4. In the section 4, line 380, there are some grammar issues in the writing for this paragraph, e.g., `been unit tested however....`, there lacks a comma before the 'however', and the content is actually no contrast in the content of this sentence.
5. Since this is a benchmark paper, it is important to know whether the author has calibrated the performance of the implemented IS methods and DR estimators? Are the performance of these methods aligned with the original research papers?
6. The equations are not fully numbered, for example, the equation about the `self-normalized weights` is not labeled, besides, it lacks explanations for this equation, and notation $\epsilon$.
7. In table 5, what does the difference mean? I.e., the difference between oracle IS vs implemented version in offline-rl-ope? or between Scope-RL vs offline-rl-ope (it seems like the two columns are duplicated, there is no further discussion regarding the table)?

---

> ### Author Response · Authors · 2024-11-21
> **Question response**
>
> Hi, thank you for taking the time to review the paper and for your comments. We completely appreciate that the paper has some layout/grammar/incompleteness issues and we will address these. With respect to your questions:
>
> __The format for the caption in Figure 1 seems not aligned, it is suggested to adjust for better presentation.__: We will address this
>
> __Figure 2 is not completed. The caption content is not described at all.__: We will address this
>
> __In line 794, what does ?? refers to?__: This was a citation that has not properly rendered - we will address this
>
> __In the section 4, line 380, there are some grammar issues in the writing for this paragraph, e.g., been unit tested however...., there lacks a comma before the 'however', and the content is actually no contrast in the content of this sentence.__: We will address this
>
> __Since this is a benchmark paper, it is important to know whether the author has calibrated the performance of the implemented IS methods and DR estimators? Are the performance of these methods aligned with the original research papers?__: With respect to benchmarking, we took the following approach:
> - Unit testing - we felt this was in essense benchmarking against the original papers since the estimators were implemented without the complications of the abstractions of the main package;
> - Benchmarking against Scope_RL for discrete action spaces;
> - Benchmarking marking for continuous action spaces was more challenging since we could not find a ground truth implementation. We could try benchmarking against the COBS library?
>
> __The equations are not fully numbered, for example, the equation about the self-normalized weights is not labeled, besides, it lacks explanations for this equation, and notation__: We will address this
>
> __In table 5, what does the difference mean? I.e., the difference between oracle IS vs implemented version in offline-rl-ope? or between Scope-RL vs offline-rl-ope (it seems like the two columns are duplicated, there is no further discussion regarding the table)?__: The difference in table 5 refers to the percentage difference in estimates between the ```offline_rl_ope``` implementation and the scope-RL implementation. The two columns refer to taking the percentage difference with respect to using the scope-RL estimate in the denominator (first column) and the ```offline_rl_ope``` estimate in the denominator (second column).
>
> __Ground truth estimates__
> - Ground truth estimates were not included since these are existing and commonly used estimators and we felt providing ground truth estimates would not necessarily be useful since the no new estimator was being proposed, rather we wanted to benchmark the implementation. That being said, we would be happy to provide them.

---

### Meta-Review · Area_Chair_T3uL · 2024-12-16

**Metareview:**

This work proposes a python package with multiple methods and metrics such as IS, WIS, PD, WPD, etc. Unfortunately the reviewers agreed that although this library seems useful, the results presented in the paper are not ready for publication. Multiple issues were raised such as "the paper was written in a rush', and the lack of clarity regarding the paper's main contributions. We encourage the authors to fix these issues in a revised manuscript.

**Additional Comments On Reviewer Discussion:**

The authors failed to convince the reviewers that their concerns regarding novelty, utility and comparison with previous work was addressed.

---

### Decision · Program_Chairs · 2025-01-22

Reject